# Enhancement of Self-Healing Efficacy of Conductive Nanocomposite Hydrogels by Polysaccharide Modifiers

**DOI:** 10.3390/polym15030516

**Published:** 2023-01-18

**Authors:** Nataša Z. Tomić, Myriam Ghodhbane, Zineb Matouk, Nujood AlShehhi, Chiara Busà

**Affiliations:** 1Self-Healing Materials Research Group, Advanced Materials Research Center (AMRC), Technology Innovation Institute (TII), Masdar City 9639, Abu Dhabi, United Arab Emirates; 2Energy Storage Group, Advanced Materials Research Center (AMRC), Technology Innovation Institute (TII), Masdar City 9639, Abu Dhabi, United Arab Emirates; 3Nano Materials Group, Advanced Materials Research Center (AMRC), Technology Innovation Institute (TII), Masdar City 9639, Abu Dhabi, United Arab Emirates

**Keywords:** conductive hydrogel, polysaccharide, PVA, reduced graphene oxide, self-healing, graphene nanoplatelets

## Abstract

The proper design of a polysaccharide/hydrocolloid modifier significantly affects the conductivity, self-healing, and viscoelastic properties of nanocomposite hydrogels. Due to the presence of different functional groups, these hydrogels can participate in the covalent, hydrogen and dynamic bonding of a system. The improvement of interactions in this system can lead to the development of high-performance nanocomposite hydrogels. In this study, resilient, self-healing and self-adhesive conductive nanocomposite hydrogels were produced by multiple and diverse coordination connections between various polysaccharide-based modifiers (Arabic gum, sodium carboxymethyl cellulose, and xanthan), the poly(vinyl alcohol) (PVA) network and different graphene-based fillers. Graphene nanoplatelets (GNP), activated carbon black (ACB), and reduced graphene oxide (rGO) have distinct functionalized surfaces, which were analyzed by X-ray photoelectron spectroscopy (XPS). Furthermore, the introduction of fillers balanced the hydrogels’ viscoelastic properties and electrical conductivity, providing the hydrogels with resilience, improved electrical conductivity, and extreme stretchability (5000%). The self-healing properties were analyzed using time-dependent measurements in a shear strain mode using an RSO Rheometer. The improvement in electrochemical and conductivity properties was confirmed by electrochemical impedance spectroscopy (EIS). The obtained conductive nanocomposite hydrogels design opens new possibilities for developing high-performance polysaccharide-based hydrogels with wearable electrical sensors and healthcare monitoring applications.

## 1. Introduction

The physical information connection between humans and their surroundings is made through the skin, which allows us to feel temperature, humidity, pressure, and other complicated environmental stimuli. Artificial skin-like materials that mimic the sensory qualities of human skin are a popular topic of study, with applications ranging from human motion detection [1], individualized health diagnostics [2], human–machine interfaces [3], and implantable biomedical devices [4]. Good conductivity, excellent stretchability, flexibility, autonomous self-healing, and self-adhesive characteristics are critical components for robotic skin-like materials. However, simultaneously designing and preparing multifunctional materials with such a wide range of applications remains difficult.

Traditional conductive materials have questionable reliability in the long term, and they are sensitive to damage during their lifetime. The enhancement of conductive materials by the self-healing properties of hydrogels has been recognized as a promising strategy to prolong their durability as sensing devices [5]. Hydrogels are three-dimensional (3D) networks of hydrophilic polymer cross-linked with varying quantities of water [6,7]. Due to their excellent biocompatibility and hydrophilic characteristics, hydrogels are investigated for various applications, including biomaterials [8] and environmental remediation [9]. Because of their molecular similarities to natural soft tissues, conductive hydrogels are among the most promising hydrogels [10] and have a wide variety of biological applications. Excellent mechanical characteristics of conductive hydrogels, such as flexibility and stretchability, are required for practical use under repetitive strain conditions. Because of its suitable mechanical properties, large surface area, biocompatibility, and flexibility, cellulose nanocrystals (CNCs) with native crystalline structures demonstrate considerable potential in boosting the mechanical performances of composites as nanofillers. Chen et al. recently reported a conductive nanocomposite organo-hydrogel with super stretchability and self-healing capabilities made from a cellulose nanofibril-based polyacrylamide (PAAm) network via boronic ester linkages [11]. Lin et al. used reversible dynamic bonds to incorporate Ag/TA@CNC nanocomposites (cellulose nanocrystals (CNCs) decorated with tannic acid (TA) and silver nanoparticles) nanocomposites into a polyvinyl alcohol (PVA) matrix to fabricate a biomimetic hydrogel with excellent mechanical strength and a fast self-healing ability [12]. TA can be also used for the introduction of hydrogen bonds in two subsequent crosslinking steps of the imine production of carboxyl methyl chitosan, oxidized cellulose nanofibers, and chitin nanofibers [13]. This conductive hydrogel had good printability, biocompatibility, and strain sensing ability. Gehong et al. synthesized supramolecular polymer network by combining conductive polyaniline (PANI) and hydrophobic association poly(acrylic acid) (HAPAA) hydrogel matrix, called (PAAN) [14]. This way, both mechanical and electrical properties were significantly improved without affecting self-healing capability.

Incorporating conductive polymers in the hydrogel matrix, such as polypyrrole [15], polythiophene [16], and polyaniline [17], is a common technique for creating conductive hydrogels. However, conductive polymers are naturally hydrophobic and incompatible with hydrophilic polymer matrixes, resulting in conductive component aggregation and nonuniform dispersion. Inadequate weak contacts between the two elements generally resulted in a poor mechanical performance of the hydrogel and poor accommodation of enormous strains, preventing realistic prospects in the field of wearable strain sensors [18]. As a result, new conductive components are required to produce conductive nanocomposite hydrogels. Therefore, nanomaterials, such as metal nanoparticles or nanowires [19], carbon-based nanomaterials [20], and MXene nanosheets [21] are used in synthesis of conductive hydrogels. Because of its superior mechanical qualities, excellent electrical conductivity, and large specific surface area, graphene is suitable for creating conductive nanocomposite hydrogels. However, because of its naturally hydrophobic characteristics and exceptional chemical durability, the homogeneous dispersion of graphene in hydrophilic hydrogel networks is difficult. The optimization of surface functionalities that enable proper interactions with the polymer matrix and conductivity is necessary. Thus, nanocomposite hydrogel is a promising material for soft bioelectronics. Intuitively conductive hydrogels (e.g., highly stretchable, elastic, and ionic conductive hydrogel for artificial soft electronics [22], on-skin paintable bio-gel for long-term high-fidelity electroencephalogram recording [23]) are attracting the attention of researchers thanks to their combination of properties. An appropriate understanding of the structure-property relationship of such systems is a critical factor in developing advanced conductive hydrogels for sensing applications.

Therefore, this study aimed to improve self-healing properties and conductivity through the addition of different polysaccharide modifiers, such as Arabic gum (AG), sodium carboxymethyl cellulose (Na-CMC), and xanthan. Conductive fillers (graphene nanoplatelets (GNP), conductive activated carbon black (ACB), and reduced graphene oxide (rGO)) were selected as they present different surface characteristics that can react differently in the systems with reversible dynamic bonding. Their structure and surface were characterized by X-ray diffraction analysis (XRD) and X-ray photoelectron spectroscopy (XPS). The size distribution of fillers was determined by Litesizer 500. Self-healing properties were quantified by time-dependent rheological behavior tested using an RSO Rheometer. Finally, electrochemical properties and conductivity were characterized by electrochemical impedance spectroscopy (EIS).

## 2. Materials and Methods

### 2.1. Materials

Activated conductive carbon black (ACB) C-NERGY Super C45 derived from TIMCAL: density = 1.86 g/cm^3^, BET surface area = 45 m^2^/g, and particle size 100–200 nm. Graphene nanoplatelet aggregates (GNP) are aggregates of sub-micron platelets with a diameter of less than 2 µm and a thickness of a few nanometers, a bulk density of 0.2 to 0.4 g/cm^3^, an oxygen content of <2 wt.%, and carbon content of >98 wt.%. BET with a surface area of 750 m^2^/g was supplied as a black powder from Strem Chemicals, Inc. A two-stage technique including oxidizing graphite to graphene oxide (GO) and then reducing it to graphene flakes was used to produce reduced graphene oxide (rGO). rGO was supplied by Advanced Graphene Products, Zielona Gora, Poland. According to the specification, rGO contains carbon: >80%, oxygen: <18%, hydrogen: <1.8%, sulfur: <0.2%; and has a flake size > 100 µm, BET specific surface area 500–700 m^2^/g, average bulk density 12 g/dm^3^, and number of layers < 7. The distance between the layers is 0.350–0.390 nm.

Poly(vinyl alcohol) (PVA) (87–90% hydrolyzed, average M_w_ = 30,000–70,000 g/mol), Arabic gum (AG) from acacia tree (branched polysaccharide of galactose, rhamnose, arabinose, and glucuronic acid as the calcium, magnesium, and potassium salts with average M_w_ = ~250,000 g/mol), sodium carboxymethyl cellulose (Na-CMC, average M_w_ = ~250,000 g/mol, degree of substitution 0.9), and sodium tetraborate decahydrate (borax) (ACS reagent, ≥99.5%) were all supplied by Sigma Aldrich, St. Louis, MO 63103, USA. Xanthan gum was food-grade (Bob’s Red Mill Natural Foods, Inc., Milwaukie, OR 97222, USA).

The common functional groups for all the used polysaccharides are hydroxyl and carboxyl groups. The mutual feature of Na-CMC and AG is that they contain salts, i.e., they are in the form of ions, and thus are expected to have a higher ionic conductivity. On the other hand, xanthan gum includes the main chain linked with a β-1,4-glycosidic bond encompassing a trisaccharide side chain of alpha-D-mannose with an acetyl group, beta-D-glucuronic acid, and a beta-D-mannose terminal unit attached to a pyruvate group [24,25].

### 2.2. Preparation of Conductive Hydrogels

The synthesis process of conductive hydrogel was performed according to the literature [26]. More precisely, two separate solutions—25 wt.% solution of PVA (2 g in 8 g of H_2_O) and 3 wt.% of polysaccharides, Na-CMC or Xan, (0.167 g in 5.33 g of H_2_O)—were prepared at 80 °C at 600 rpm. In the case of AG, 0.167 g of AG was directly added to the PVA solution since the AG solution showed a high loss in viscosity when heated. After the complete dissolution of PVA and polysaccharides, the two solutions were mixed using a mechanical stirrer at 80 °C at 600 rpm. Then, 1–5 wt.% of graphene-based fillers ACB/GNP/rGO (relative to PVA solid content) were added to the mixture and stirred for the next 10 min. Finally, a borax solution (0.12 M, 3.33 mL) was added dropwise into the prepared mixture, and by applying kneading technique, the conductive nanocomposite hydrogel was finally prepared.

Figure 1 shows a schematic illustration of the synthesis mentioned above, consisting of dissolving, mixing, and cross-linking processes.

### 2.3. Characterization Methods

The size distribution of ACB, GNP, and rGO was determined by the dynamic light scattering (DLS) method using a particle analyzer LiteSizer 500 from Anton Paar, Germany. The measurements were performed in a back-scattering configuration at 25 °C in isopropanol.

Powder X-ray diffraction (XRD) tests were recorded on a Bruker D8 Advance instrument (Bruker, Germany). Cu-Kα radiations were employed as an X-ray source (λ = 0.154 nm). The test was run in the 0–60° range at room temperature, with an 0.01° increment and 0.5 s/step.

The surface atomic composition of graphite-based fillers was examined by XPS using a PHI VersaProbe 5000 Scanning X-ray Photoelectron Spectrometer with an Mg Ka X-ray source (1100 eV). A monochromated Mg X-ray source (1100 eV) was used as a probe for the experiments. The X-ray beam power was 50.17 W with the step size of 0.05 eV and detector pass energy of 280 eV. An E-neutralizer (1V) and I-neutralizer (0.11 kV Ar^+^ ion) were implemented during the experiment. The compositions were calculated by using the area under the high-resolution curve and weighted with the respective sensitivity factors for each elemental species. Peaks were calibrated using the C1s as a reference at 284.6 eV. The software Casa Xps was used for curve fitting and calibration.

The viscoelastic properties of conductive nanocomposite hydrogels were analyzed on a rheometer at 25 °C (Brookfield RSO oscillatory Rheometer supplied by Venktron, UAE) in conical plate mode. Amplitude sweep measurements were performed in a shear strain mode with a start amplitude of 0.06%, end amplitude of 200.00%, frequency of 10 Hz, and for 30 steps; they showed the yield and flow point of hydrogels. Time-dependent behavior was evaluated in 5 blocks in a shear strain mode, alternating the amplitude of 1% and 200% at 10 Hz during 60 s for each block. Frequency Sweep measurements were also performed in a shear strain mode with 0.10% amplitude, starting from 0.10 Hz until 10 Hz and 30 steps.

Electrochemical impedance spectra (EIS) were recorded in the frequency range of 100 kHz−10 mHz with a potential amplitude of 10 mV on BioLogic’s VSP-300 channel potentiostat.

## 3. Results

Considering the unstable nature of the surface of graphene-based fillers, the appropriate characterization and comparisons were performed by a particle size analyzer Litesizer 500, XRD, and XPS. These methods provide a comprehensive and detailed analysis of surface characterization, which will enable a better understanding of interactions in conductive nanocomposite hydrogels.

Figure 2 shows particle size distribution and XRD diffractograms. The lowest diameter was found for GNP particles (71.4 nm), where the second peak can be attributed to the appearance of agglomerates in isopropanol, which is expected due to the high affinity of nanoparticles to form aggregates [27,28,29] due to the weak dispersibility of graphene on isopropanol [30]. According to the manufacturer’s specification, the particle diameter size of ACB particles falls into the expected range of 100–200 nm. According to the manufacturer, rGO should have a flake size > 100 µm, but it was measured at 200–600 nm with a peak at 360.8 nm. All the used particles had larger diameters than human skin pores, where no penetration was expected [31].

In order to gain a better understanding of the chemical bonding between carbon and oxygen functionalities, high-resolution XPS of the C 1 s peak of both rGO and GNP samples was deconvoluted into three chemically shifted segments at binding energies of 284.6, 286, and 288.2 eV, respectively, Figure 3b [32,33].

The non-oxygenated carbon of C–C/C–H, representing the graphene structure, was designated as the first peak C1 [34]. The XRD measurement corroborates this claim, as shown in Table 1. The interaction of carbon atoms with oxygen in either hydroxyl (C–OH) or epoxide (C–O) functional groups are responsible for the second peak C2 at 286 eV. Carbon functionalized C3 in COOH is related to the carbonyl (>C=O) peak. The intensity of the carbonyl peak was a bit higher in GNP than in the rGO samples, indicating a higher content of COOH complexes. This functional group is often found at the edges of the graphene sheets, while the epoxy group evolves at the basal plane of the graphene sheets, resulting in in-plane defects and disorders [35]. The ratio of the C1 peak intensity to the sum of the (C2 + C3) peak intensities was smaller in rGO (1.3) than in GNP (4), indicating a higher content of oxygen functionalities in rGO. This result suggests that the rGO sample has a denser epoxy network at the basal plane and a higher quantity of carbonyl connected at the edges of the graphene sheets than GNP, which could be related to the rGO sample’s harsher oxidation reaction [36]. ACB showed only C1 and C2 peaks with higher oxygen contents.

Lower oxygen values can suggest a higher conductivity but also incompatibility with hydrogels, and thus the appearance of aggregates. Presented ratios can indicate the better compatibility of ACB and rGO with hydrogels due to higher oxidation levels.

The presence of hydroxyl groups in the structure of ACB and rGO can also participate in cross-linking reactions with borax and contribute to improved self-healing efficacy. The presence of carboxyl groups, on the other hand, in the structure of rGO can take part in the esterification reaction during the hydrogel synthesis. Higher covalent bonding, such as ester bonds, can improve the mechanical properties of a hydrogel.

To evaluate the self-healing properties of synthesized hydrogels, the amplitude sweep measurements were first performed in a shear strain mode (0.06–200.00%, 10 Hz). The obtained results of the yield and flow point of hydrogels are presented in Figure 4. In a low strain region, both the storage modulus (G′) and loss modulus (G″) remain constant values up to the yield point. The yield point for PVA represents the reduction in G values. At the same time, for all other samples with a polysaccharide modifier, this was the value at which an increase in storage modulus occurred.

Nevertheless, all the tested samples exhibited a significant decrease in G′ values for strains higher than 1%. The flow point represents the value of intersections of G′ and G″ curves when the hydrogel network starts to flow. According to the results presented in Table 2, the flow of hydrogels is higher, indicating improved self-healing properties. With the further increase in strain, the values significantly decrease due to the disintegration and collapse of the hydrogel network.

Thus, the time-dependent behavior of hydrogels was evaluated in five blocks in a shear strain mode, alternating the amplitude of 1% and 200%, as shown in Figure 4. This measurement gave an insight into the self-healing efficacy since the 200% strain caused hydrogels to disintegrate, and then 1% enabled the self-healing and recovery of viscoelastic properties. Neat PVA hydrogel showed inconsistent self-healing results, i.e., the first recovery was even higher than the initial recovery, but the later recovery was lower than the first one, Figure 4a. Moreover, while performing the test, it was observed that some parts of PVA hydrogel went out of the disc surface and could not integrate again with the bulk material. Similar G recovery variations were found for PVA/CMC hydrogel, Figure 4c. The PVA/AG hydrogel exhibited the highest initial values, but they became lower each time after a high strain cycle, Figure 4b. The most consistent values and 100% recovery of G′ values were found for PVA/Xan sample. The reasons for the superior self-healing properties of PVA hydrogel due to the addition of xanthan gum as a polysaccharide modifier are its branched structure, functional groups that participate in dynamic bonding, and its shear-thinning properties [37]. Since the self-healing capability was one of the most important features of these materials, electrical properties were more focused on composite hydrogels based on PVA/Xan.

The influence of different polysaccharide modifiers and filler of PVA hydrogels on rheological properties in a frequency sweep test is presented in Figure 5. The storage modulus (G′) represents the hydrogel strength [26], and this is affected by all the changing parameters (constituents and ratios in composition). Figure 5a shows similar final values of G′ at 10 Hz but much lower values for PVA/CMC. The same case was found for the loss modulus (G″). Figure 5b shows the positive impact of ACB addition on PVA/Xan hydrogel, where the increase in G′ at 10 Hz compared to neat PVA/Xan was 59.6% for 1 wt.%, 99.2% for 3 wt.%, and 127.6% for 5 wt.% of ACB. A smaller difference is observed for the G″ values. All types of fillers (ACB, GNP, and rGO) improved the G′ values of neat PVA/Xan hydrogel, as shown in Figure 5. One observed phenomenon is that rGO significantly improves rheological properties at low frequencies, but at higher frequencies, such as 10 Hz, it becomes lower than both ACB and GNP but still higher than neat PVA/Xan. Compared to the literature, G′ obtained in this study is more than 30 times higher and performed at the same rate of 10 Hz [26].

One of the potential applications of synthesized hydrogels is artificial skin to simulate human skin abilities. Figure 6a represents hydrogels as the tip of a touch screen pen that imitates the human finger. This property indicates the potential in robotics, flexible electronics, and human–machine interactions. The visual presentation of self-healing capability without external stimuli is shown in Figure 6b. Two different hydrogels were rejoined for better visualization, neat PVA, and the PVA/Xan + 5 wt.% of ACB. By manual stretching, the effective rejoining becomes a significant and flowing phenomenon. According to Table 2, the highest value for a flow point (%) was found for PVA, and one of the lowest values was found for PVA/Xan + 5 wt.% ACB (3.6 times higher). Therefore, the stretching presented in Figure 6b represents a ~3 times higher stretchability of PVA/Xan + 5 wt.% ACB sample compared to neat PVA.

As shown in Figure 6c, we created an LED circuit to visually demonstrate the electrical healing process of the synthesized hydrogels. To maintain electrical conductivity, the red LED bulb was first lighted in the circuit connected by the conductive hydrogel flattened on a copper substrate using a commercial lithium battery of 3V. The LED bulb light turned off when the conductive hydrogel was split into two parts. After recontacting the two broken parts, the circuit was restored with the healed conductive hydrogel, and the LED bulb was instantaneously lighted.

The stretching capability and flexibility of obtained conductive nanocomposite hydrogels were very appealing, and thus, these hydrogels were further tested, as shown in Figure 7. A stretchability of ~5000% was obtained, which is the highest value obtained in the literature for conductive nanocomposite hydrogels [38,39,40].

Aside from extremely high stretchability, the synthesized hydrogels are moldable to any necessary shape and any required size, as shown in Figure 8a. The flowability of a conductive hydrogel based on PVA/Xan enabled good contact with various surfaces such as the skin, as shown in Figure 8b. The removed hydrogel from the skin showed inverse skin surface morphology, as shown in Figure 8c. Self-adhesiveness ability was also tested on other substrates such as glass, as shown in Figure 8d.

To gain an insight into the conductivity of synthesized hydrogels, the electrochemical impedance spectroscopy (EIS) with different polysaccharide modifiers, fillers, and amounts of fillers was performed, Figure 9. These results show that PVA and PVA/Xan have similar conductivity values, while PVA/AG and PVA/CMC showed 16.3% and 33.4% higher values than PVA, respectively. The reason for this lies in the fact that both polysaccharide modifiers AG and CMC possess ionic conductivity as their structure includes salts. Na-CMC has a higher content.

The conductivity in the function of the added amount of ACB filler is presented in Figure 9d. The values for 1 wt.% and 3 wt.% of added ACB are much higher than neat PVA/Xan, for 17.7% and 16.7%, respectively. The absence of improvement by the addition of 3 wt.% ACB may be attributed to aggregates since no homogenization technique has been used, as demonstrated in the literature [26]. Nevertheless, the conductivity further increased with further addition of ACB since it was 32.3% higher than PVA/Xan in the case of the PVA/Xan + 5 wt.%. Figure 9a and b shows Nyquist diagrams for neat PVA/Xan and conductive nanocomposite hydrogels with 5 wt.% of different fillers (GNP, ACB, and rGO). The highest conductivity (S) was found for rGO fillers. S for rGO fillers was 29.8% and 6.5% higher than GNP and ACB, respectively. Very similar electrochemical properties of ACB and rGO suggest a high potential for using ACB as a cost-effective and conductive filler. It is worth mentioning that the conductivity results obtained in this study are significantly higher than those for PVA-modified tannic acid-decorated cellulose nanocrystals with the addition of partially synthesized rGO [26].

## 4. Conclusions

This study designed and presented a conductive nanocomposite hydrogel with extreme stretchability, a rapid self-healing capability, self-adhesive properties, and excellent adherence to different substrates. Due to the presence of polysaccharide modifiers in PVA-based hydrogels, the optimization of properties of conductive hydrogels was achievable. Xanthan gum contributed to the complete recovery of rheological and self-healing properties. The reasons for the best self-healing properties of PVA hydrogels by the addition of xanthan gum as a polysaccharide modifier are its branched structure, functional groups that participate in dynamic bonding, and its shear thinning properties. Arabic gum and sodium carboxymethyl cellulose improve conductivity due to their ionic structure. Surface functionalities of fillers that could participate in reversible dynamic bonding contributed to higher conductivity, such as rGO and ACB. Finally, this hydrogel design presents a unique technique for manufacturing high-performance hydrogels with tremendous potential as wearable bioelectronics.

## Figures and Tables

**Figure 1 polymers-15-00516-f001:**
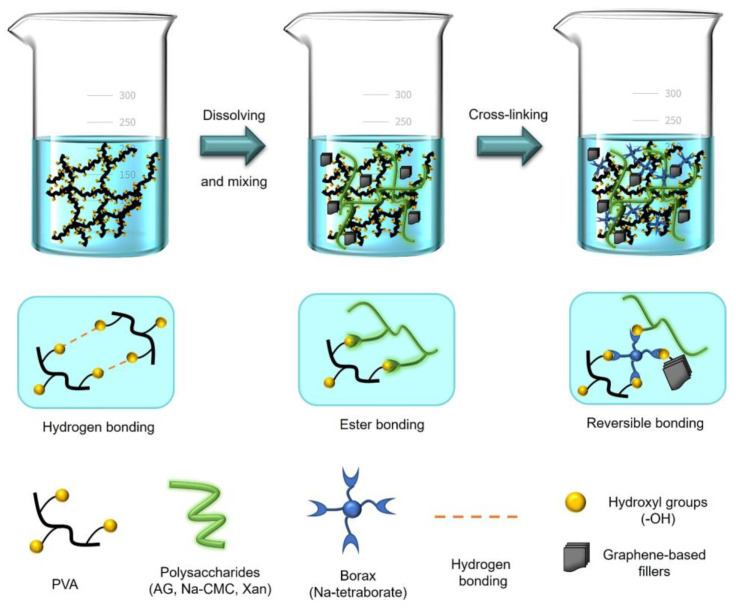
Schematic illustration of the synthesis process of conductive nanocomposite hydrogels.

**Figure 2 polymers-15-00516-f002:**
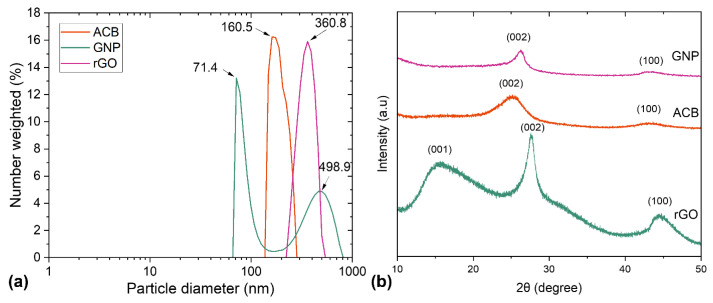
Characterization results of used fillers ACB, GNP, and rGO by (**a**) particle size analyzer, and (**b**) XRD patterns.

**Figure 3 polymers-15-00516-f003:**
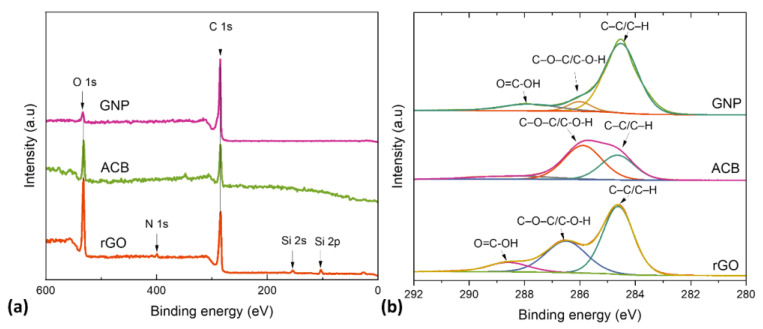
Chemical structure characterization by XPS: (**a**) Survey spectra; (**b**) XPS fitted spectra of C (1S) of rGO, ACB and GNP powder.

**Figure 4 polymers-15-00516-f004:**
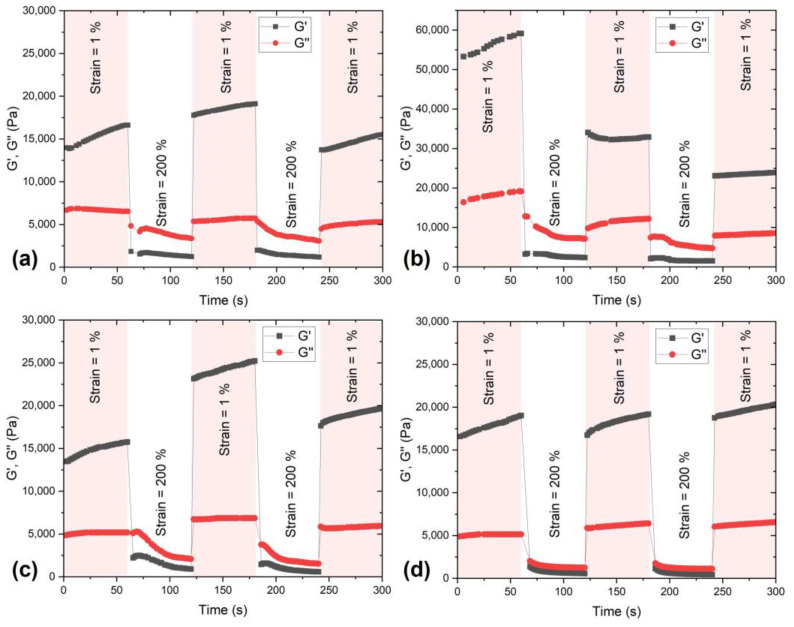
Self-healing efficiency evaluated by alternating the low (1%) and high (200%) shear strain of hydrogels: (**a**) PVA, (**b**) PVA/AG, (**c**) PVA/CMC, and (**d**) PVA/Xan.

**Figure 5 polymers-15-00516-f005:**
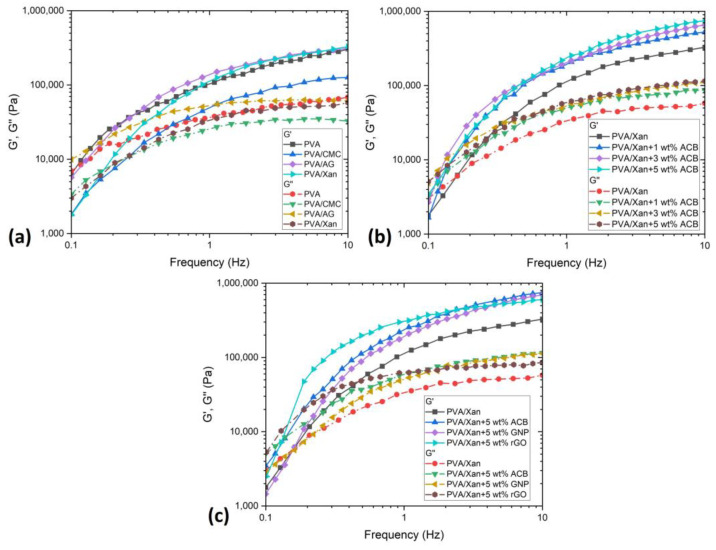
Viscoelastic properties of hydrogels with a dependence on the frequency sweep of hydrogels with (**a**) different polysaccharide modifiers, (**b**) different percentages of fillers, and (**c**) different fillers.

**Figure 6 polymers-15-00516-f006:**
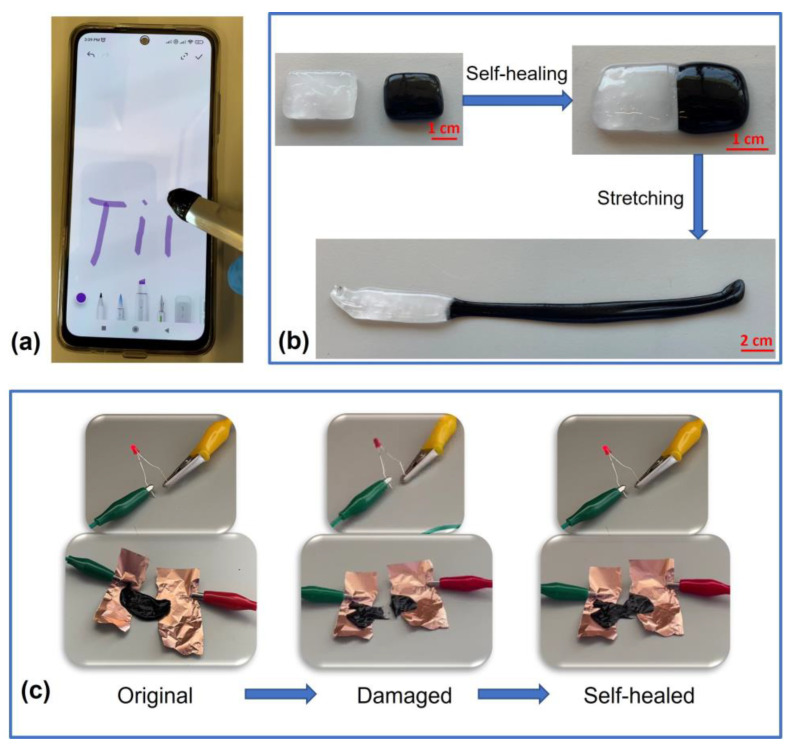
Illustration of the applications and the self-healing capability of conductive hydrogels: (**a**) touch screen property, (**b**) self-healing by rejoining without any external stimuli, and (**c**) the regaining of the conductive properties after the self-healing process.

**Figure 7 polymers-15-00516-f007:**
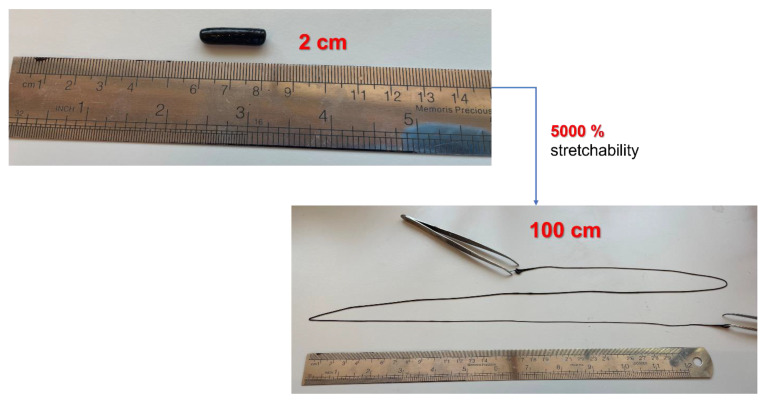
Extreme stretchability observed for the sample PVA/Xan + 5 wt.% ACB.

**Figure 8 polymers-15-00516-f008:**
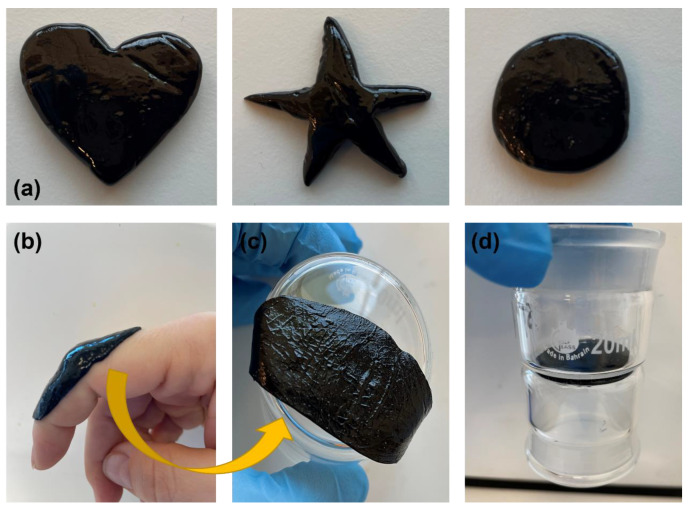
(**a**) Moldability of samples into different shapes, (**b**) self-adhesion on skin, (**c**) morphology of the hydrogel contact surface with the skin, and (**d**) example of adhesion to glass.

**Figure 9 polymers-15-00516-f009:**
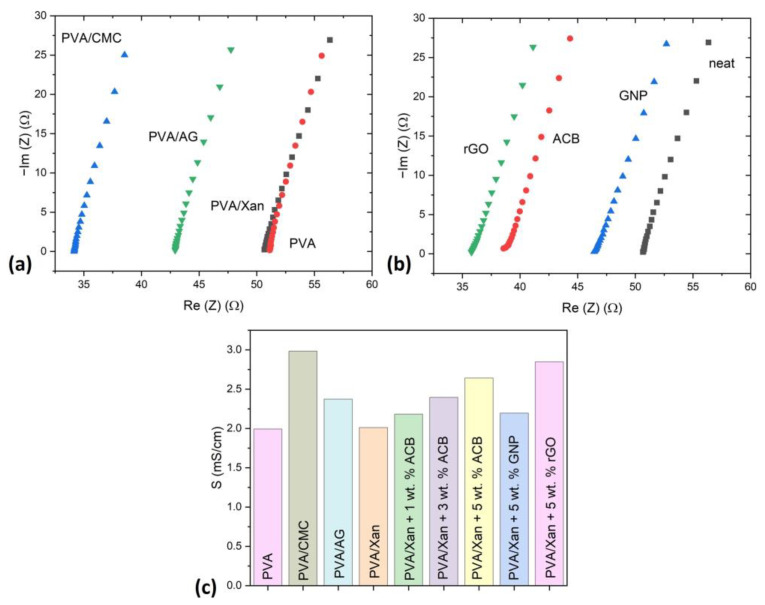
EIS measurements: (**a**) Nyquist diagram for hydrogels with different polysaccharide modifiers, (**b**) Nyquist diagram for PVA/Xan with 5 wt.% of different fillers and (**c**) conductivity for all samples.

**Table 1 polymers-15-00516-t001:** Chemical composition (atomic %) determined using XPS.

	C, %	O, %	N, %	Si, %	C1 (C-C/C-H), %	C2 (C-O-C/C-O-H), %	C3 (O=C-OH), %	C1/C2 + C3, %
Binging energy, eV	285	532	399	99	284.6	286.4	288.9	/
ACB, %	73.9	26.1	/	/	40	60	/	0.7
rGO, %	65.3	29.1	1.7	3.8	56	34	10	1.3
GNP, %	95.5	4.5	/	/	79.5	7.6	12.9	4

**Table 2 polymers-15-00516-t002:** Yield and flow point values from strain-dependent oscillatory shear rheology of the PVA based hydrogels at a fixed frequency (10 Hz).

Sample	Yield Point, %	Flow Point, %
PVA	4.00	98.17
PVA/AG	0.42	17.84
PVA/CMC	0.08	31.18
PVA/Xan	0.30	25.15
PVA/Xan + 1 wt.% ACB	0.22	30.18
PVA/Xan + 3 wt.% ACB	0.10	27.51
PVA/Xan + 5 wt.% ACB	0.08	26.91
PVA/Xan + 5 wt.% GNP	0.06	22.34
PVA/Xan + 5 wt.% rGO	0.06	25.09

## Data Availability

Not applicable.

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
