# Peer review of "Enhancement of Self-Healing Efficacy of Conductive Nanocomposite Hydrogels by Polysaccharide Modifiers"

_polymers, 2023, doi:10.3390/polym15030516_

Round 1

Reviewer 1 Report

Overall, the improvement in conductivity is a promising outcome.  The language could be revised to be more of academic writing e.g. using “coworkers” is not suitable, “et al” should have been used. Introduction needs to be revised to include all the progress in the development of self-healing hydrogel soft sensors, and justify the need for the current study.

The self-healing properties and conductivity of different modifier have been demonstrated, however critical analyzing of the results are mainly missed. The article should include a discussion to compare the results to those reported by other groups[1-3] on advancements in the development of self-healing hydrogel soft sensors. 

1.         Heidarian, P., et al., A 3D printable dynamic nanocellulose/nanochitin self-healing hydrogel and soft strain sensor. Carbohydrate Polymers, 2022. 291: p. 119545.

2.         Su, G., et al., Balancing the mechanical, electronic, and self-healing properties in conductive self-healing hydrogel for wearable sensor applications. Materials Horizons, 2021. 8(6): p. 1795-1804.

3.         Qin, T., et al., Recent progress in conductive self-healing hydrogels for flexible sensors. Journal of Polymer Science, 2022. 60(18): p. 2607-2634.

Author Response

First, the authors would like to thank the Reviewer and Editor for their valuable comments, which helped us revise the manuscript and present it in a much better form. Here are responses that follow your comments, and the Track Changes command shows all the changes in the manuscript.

Reviewer 2 Report

This manuscript presents several kinds of PVA hydrogel incorporated with different polysaccharide modified carbon materials and evaluated their mechanical, conductive, self-healing and self-adhesive properties. Unfortunately, after carefully reviewing the manuscript, I think the results descripted in this manuscript is not suitable for publication in the journal of Polymers. The comment is listed as follows.

Classical nanocomposite hydrogels are hydrogels crosslinked by nanoparticles and usually possess excellent mechanical properties. However, in this paper, the nano-size modified carbon materials are merely mixed with the PVA solution. The cross-links were contributed the weak interactions between the function groups on the polysaccharide and PVA. Although the borax was used to crosslinked the PVA chain, no evidence was provided that the cross-links were really formed. Therefore, the prepared materials reveal very weak mechanical properties, as described in Line 214G' values for 214strains higher than 1%. Furthermore, as can be seen from Figure 6 and Figure 7, the prepared materials should be defined as PVA pastes, instead of hydrogels. which are nearly useless for hydrogel application. As a kind of paste, it is not strange that it can be stretched, and merged together and adhesive to skin and other surfaces. Therefore, I suggest this paper to submit to other journal after revising the title and parts of its content.  

Author Response

First, the authors would like to thank the Reviewer for his valuable comments.

The authors would like to mention that the aim of this manuscript was not to improve mechanical properties. Actually, only the viscoelastic properties are measured by the RSO Rheometer. This technique was used to confirm and compare the self-healing capability of tested hydrogels. The purpose of using graphene-based fillers was to improve self-healing and stretchability (shown in this study) but also to improve sensitivity (part of the next study/paper).

The cross-linking evidence is that the sample got a solid form and that it was mouldable (shown in Figures within the manuscript). This phenomenon is proven and accepted by the wide scientific community, especially for PVA hydrogels crosslinked with borax. Opposed to hydrogels, the pastes cannot be handled easily and cannot maintain their shape, since they have more liquid viscosity. This is also proven with the RSO rheometer. All the hydrogels had both Yield and Flow points, which means that only when the Flow point is reached, the hydrogels started to flow. 

Round 2

Reviewer 1 Report

Thanks for providing the revision. Unfortunately the major revision has not been address. The article should include a discussion section to compare the results to those reported by other groups such as [1-3] on advancements in the development of self-healing hydrogel soft sensors. without a proper discussion the claims in the paper are not justified adequately.  

1.         Heidarian, P., et al., A 3D printable dynamic nanocellulose/nanochitin self-healing hydrogel and soft strain sensor. Carbohydrate Polymers, 2022. 291: p. 119545.

2.         Su, G., et al., Balancing the mechanical, electronic, and self-healing properties in conductive self-healing hydrogel for wearable sensor applications. Materials Horizons, 2021. 8(6): p. 1795-1804.

3.         Qin, T., et al., Recent progress in conductive self-healing hydrogels for flexible sensors. Journal of Polymer Science, 2022. 60(18): p. 2607-2634.

Author Response

The authors would like to thank the reviewer for the suggestion and the opportunity to update the manuscript. Now, the revised manuscript contains mentioned discussion and references.

Reviewer 2 Report

The authors have improved the manucript according to the comments. I agree this version to be publised in the journal of Polymers.

Author Response

The authors would like to thank the reviewer for their comments and suggestions that helped us to present our manuscript in a much better way.

Round 3

Reviewer 1 Report

The manuscript is adequately improved, Thank you for addressing the revisions. 

Author Response

The authors would like to thank the comments of the Academic Editor, and we hope that you will find our manuscript now appropriate for publishing. The authors added mentioned discussion in point No. 4.